# Research

cognition, computational biology, neuroscience

vision, colour constancy, colour relations, natural scenes

**Author for correspondence:**
David H. Foster
e-mail: d.h.foster@manchester.ac.uk

# Colour constancy failures expected in colourful environments

David H. Foster[1] and Adam Reeves[2]

[1]Department of Electrical and Electronic Engineering, University of Manchester, Manchester, UK
[2]Department of Psychology, Northeastern University, Boston, MA, USA

DHF, 0000-0003-2428-715X; AR, 0000-0003-2238-6349

Colour constancy refers to the constant perceived or apparent colour of a surface despite changes in illumination spectrum. Laboratory measurements have often found it imperfect. The aim here was to estimate the frequency of constancy failures in natural outdoor environments and relate them to colorimetric surface properties. A computational analysis was performed with 50 hyperspectral reflectance images of outdoor scenes undergoing simulated daylight changes. For a chromatically adapted observer, estimated colour appearance changed noticeably for at least 5% of the surface area in 60% of scenes, and at least 10% of the surface area in 44% of scenes. Somewhat higher frequencies were found for estimated changes in perceived colour relations represented by spatial ratios of cone-photoreceptor excitations. These estimated changes correlated with surface chroma and saturation. Outdoors, the colour constancy of some individual surfaces seems likely to fail, particularly if those surfaces are colourful.

## 1. Introduction

Ideally, for an object or surface to be recognizable by its colour, it should appear or be perceived as the same independent of the accident of illumination, whether, for example, from a blue sky or a yellow-orange sun. But in practice, the extent of this colour constancy with changes in the intensity and spectral composition of the illumination has been found to vary [1–4]. Based on more than 60 measurements from different laboratories, the most common report has been of human observers' performance falling short, with a level of constancy of around 0.74 on a scale of 0 to 1, where 0 is no constancy and 1 is perfect constancy [2,5].

These failures in colour constancy have been attributed to the methods of measurement, the incompleteness of sensory adaptation, and the instructions given to observers or how they are interpreted. Only occasionally have the colorimetric properties of the surfaces under test been considered. Yet some coloured surfaces have long been known to change their appearance, in brightness or hue, as the illumination colour changes, even when the eye is allowed to adapt [6,7]. Analyses of Munsell paint samples [8] and model reflectance spectra [9], and theoretical arguments [10], have suggested that departures from constancy should increase as colourfulness increases. Experimental measurements with mosaic and chequerboard displays of coloured surfaces made from Munsell reflectances have offered some support for these claims [11,12], but have not been extended to real-world scenes, with their different colour gamuts.

What do constancy failures with natural surfaces look like? Figure 1 shows colour images rendered from a hyperspectral reflectance image of a garden scene [14] under different simulated daylight illuminants: on the left, a mixture of skylight and sunlight representing average daylight with a correlated colour temperature (CCT) of 6500 K, and on the right, light from the setting sun with a CCT of 4000 K. The square patches around the images are enlarged copies of the pixels indicated by the connecting lines. The small neutral sphere near the top of the scene reveals the colour of the illuminant. Sample spectra from the scene are given in [14].

*Proc. R. Soc. B* **289**: 20212483

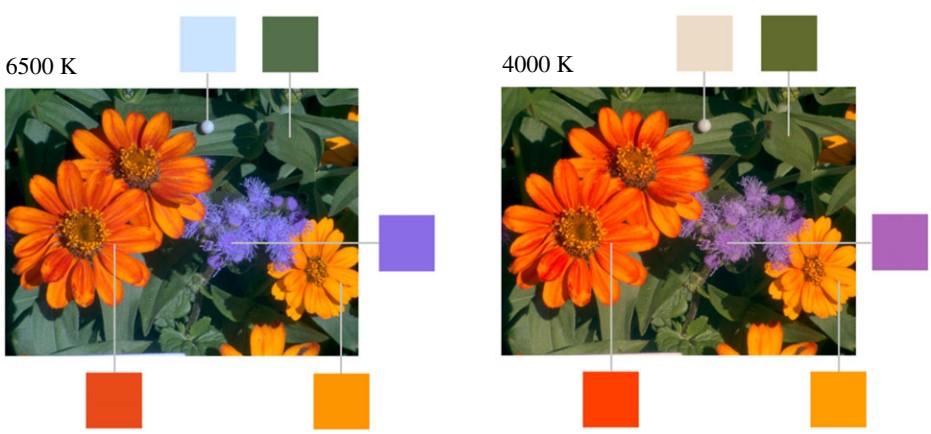

**Figure 1.** Colour images of a scene under different simulated daylights. The images are rendered as sRGB images [13] from a hyperspectral reflectance image [14] with a daylight illuminant of correlated colour temperature of 6500 K in the left panel and 4000 K in the right panel. The square patches are enlarged copies of the indicated pixels. (Online version in colour.)

The colour appearance of the surfaces is clearly different in the two conditions. The sphere appears predictably bluish under the 6500 K illuminant and orangish under the 4000 K illuminant, the purple flower appears redder, and the orange flower on the left of the scene a little lighter. Viewed successively rather than side-by-side, and with an adapted eye, the sphere should appear much the same, but the purple and orange flowers are still likely to differ [6,7].

There are parallel changes in the perceived relations between surface colours [15–18]. Most obviously, the hue difference between the purple and orange flowers under the 6500 K illuminant appears smaller under the 4000 K illuminant, an effect which is largely independent of adaptation. This kind of change has been taken as a failure of relational colour constancy [15,19]

The foregoing considerations prompt two basic questions about colour constancy in the real world. First, are failures of constancy rare? Second, if they do occur within a scene, are they correlated with measures of colourfulness? After all, most outdoor scenes do not consist of orange and purple flowers [20–25], nor do their gamuts span the spaces of Munsell or model reflectance spectra mentioned earlier.

It is difficult to address these questions by asking observers to match surfaces outdoors under changes in natural illumination. In addition to the challenge of sampling and adjusting stimuli in these conditions, the continuous variation in the spectral and spatial distribution of the illumination makes perceptual measurements difficult [26,27]. Unsurprisingly, there have been few attempts at this task [28].

Here, instead, a computational approach was taken using data from 50 hyperspectral reflectance images of natural outdoor scenes, as illustrated in figure 2. The analysis drew on reference data from previous psychophysical measurements [29]. The results suggest that outdoor environments may well contain individual colourful surfaces or parts of surfaces that fail to be colour constant.

## 2. Methods

### (a) Hyperspectral reflectance data

Scene data were taken from a set of 50 hyperspectral images of outdoor scenes drawn from the main land-cover classes [30,31], either predominantly vegetated, containing woodland, shrubland, herbaceous vegetation (e.g. grasses, ferns, flowers) and cultivated land (fields), or predominantly non-vegetated, containing barren land (e.g. rock), urban development (residential and commercial buildings), as well as farm outbuildings, and painted or treated surfaces. They were acquired from the Minho region of Portugal in 2002 and 2003 and are similar to those used in other studies [32,33]. Details of image acquisition and calibration are described elsewhere [14,32]. Each image had dimensions approximately 1344 × 1024 pixels, corresponding to a camera angle of approximately 6.9° × 5.3°, and spectral range 400–720 nm sampled at 10 nm intervals.

Each image was processed as an effective spectral reflectance image $R(\xi, \eta; \lambda)$, indexed by spatial coordinates $\xi$, $\eta$ and wavelength $\lambda$ (notation differs a little from [14]). The viewing geometry was assumed fixed and orientation dependence is therefore suppressed here. The reflectance is effective in the sense of being derived by spectrally scaling the radiance image by the reflectance of one or more reference surfaces embedded in the scene [14]. Because surfaces oriented at an angle to the camera may reflect more light than the reference surface, values of $R(\xi, \eta; \lambda)$ sometimes exceeded unity. If so, the whole image was scaled by its maximum. By definition $R(\xi, \eta; \lambda)$ confounds reflectance and spatial variation in illumination, such as from direct to indirect illumination (typically shade), but the confound may be eliminated as explained in the following section. A separate control calculation demonstrated that results are similar whether points with low reflectance values, usually in darker, shadowed image areas, are included or not.

### (b) Scene illuminant changes

Illumination changes were simulated with spectral changes in a spatially uniform, global illuminant $E(\lambda)$. This device ensures that equal changes in illumination spectrum occur at each point in the scene, isolating the role of surface reflecting properties [32]. The alternative of simulating the spatial and spectral variation of natural illumination changes due to cloud and solar elevation [26,27] introduces different spectral changes at each point, which complicates the interpretation of reflecting properties. As to the use of an effective spectral reflectance $R(\xi, \eta; \lambda)$, multiplying $R(\xi, \eta; \lambda)$ at a point $(\xi, \eta)$ by successive spectra $E(\lambda)$ is equivalent to multiplying the true local spectral reflectance at that point by successive local illumination spectra undergoing the same spectral change. Details are given in the electronic supplementary material.

The illuminants $E(\lambda)$ were drawn from the CIE daylight spectral distributions [34] with CCTs in the test condition of 4000 K and 25 000 K and in the reference condition 6500 K. Recall that

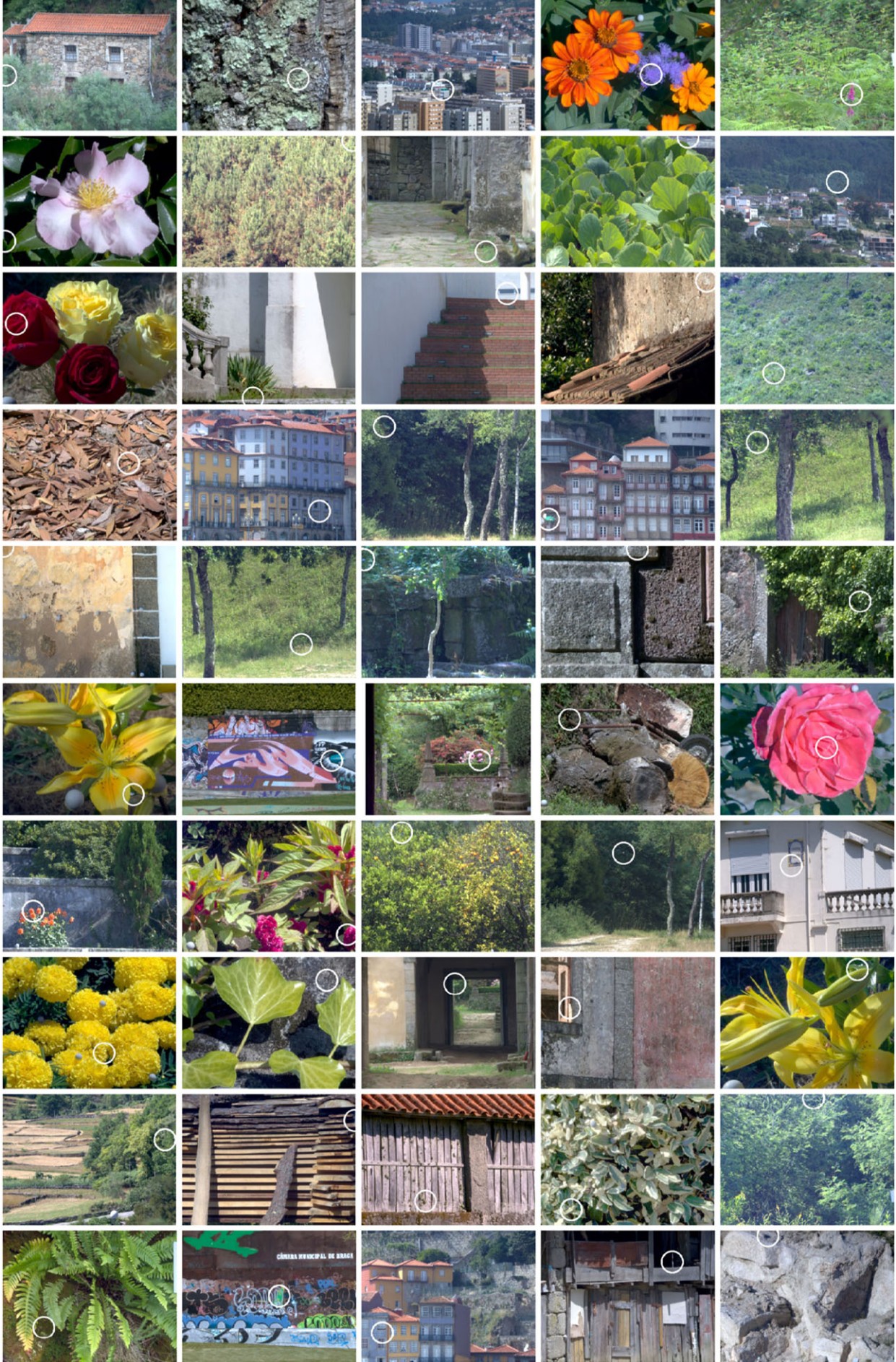

**Figure 2.** Colour images of the 50 scenes used in the analysis. Each image is rendered as an sRGB image from a hyperspectral reflectance image with a daylight illuminant of correlated colour temperature 6500 K. White circles show the location of the largest estimated colour difference under changes in daylight illuminant (some locations appear dark in these images). (Online version in colour.)

the 4000 K illuminant represents light from the setting sun and the 6500 K illuminant average daylight. The 25 000 K illuminant represents light from the north or polar sky. On a reciprocal colour temperature scale, 6500 K falls roughly midway between 4000 K and 25 000 K.

## (c) Image sampling

Reflectance images were downsampled by spatial averaging over $4 \times 4$ pixels in order to reduce non-imaging noise in the unaveraged source data [14] and pixel-pixel correlations with the 1.3-pixel line-spread function of the hyperspectral camera [32]. In what follows, points sampled from scenes refer to these $4 \times 4$-pixel surface elements. To further reduce noise, images were smoothed spectrally by averaging over adjacent wavelengths. Estimates of expected colour errors were made with and without points with low reflectance values (less than 10% over 420–700 nm).

This point-based sampling approach is neutral with respect to physical scene structure [32]. No attempt was made to associate the effective reflectance at each point with an extended physical object, whose bidirectional reflectance distribution function may itself be difficult to define [35,36].

## (d) Estimating colour differences

Expected changes in colour appearance were estimated with formulae standardized by the Commission Internationale de l'Eclairage (CIE). These formulae were used in the same practical way as in psychophysical experiments where observers discriminated between images of natural scenes [29], that is, as a physical measure of colour differences rather than as a model of the underlying perceptual processes [37].

At each point $(\xi, \eta)$ in the scene, the reflected radiance spectrum $L(\xi, \eta; \lambda)$ was converted into CIE XYZ tristimulus values, which were then mapped into an approximately uniform colour space. Details of this procedure, including the approximations involved, are described in [14]. The observer was assumed to be chromatically fully adapted to each illuminant [38,39]. Although intended to minimize colour differences, adaptation does not itself imply that colour constancy should be perfect at the level of cone photoreceptors. The reason is that given a reflectance image $R(\xi, \eta; \lambda)$ and a change in illuminant from $E_1(\lambda)$ to $E_2(\lambda)$, there is no transformation guaranteed to convert the cone response to one radiance image $E_1(\lambda)R(\xi, \eta; \lambda)$ into the response to the other $E_2(\lambda)R(\xi, \eta; \lambda)$ [40].

For generality, two approximately uniform colour spaces were used, each with an associated colour difference metric $\Delta E$ (the same symbol is used for both and should not be confused with $E(\lambda)$ used for illuminant spectra). One of the two colour spaces was CIELAB space [34], which has coordinates $L^*$, $a^*$, $b^*$, where $L^*$ represents lightness and $a^*$ and $b^*$ represent redness–greenness and yellowness–blueness, respectively, each derived from the corresponding CIE XYZ tristimulus coordinates. Let $\Delta L^*$, $\Delta a^*$, $\Delta b^*$ be the differences in $L^*$, $a^*$, $b^*$ values produced by a particular pair of illuminants. As illustrated by figure 1, changes can take the form of chromatic differences or lightness differences. The total colour difference [41] at a point is given by $\Delta E = [(\Delta L^*)^2 + (\Delta a^*)^2 + (\Delta b^*)^2]^{1/2}$. Since CIELAB space does not tolerate illuminants very different from average daylight [34,42], a result of what has sometimes been called the wrong von Kries transformation [43], a conventional chromatic adaptation transform CMCCAT2000 [44] was introduced before the calculation of colour differences. The latter can also be improved, but for compatibility with previously published work, no more adjustments were made.

The other colour space was CAM02-UCS [34], which has coordinates $J'$, $a'_M$, $b'_M$, analogous to $L^*$, $a^*$, $b^*$ of CIELAB space. Colour differences $\Delta E$ were calculated in the same way as for CIELAB space. The uniformity of CAM02-UCS is better than that of CIELAB space, though still not perfect [45]. Its inbuilt chromatic adaptation transform is modified from CMCCAT2000 [46].

Estimated colour differences $\Delta E$ were averaged over illuminant changes from 4000 K to 6500 K and 25 000 K to 6500 K and then compared with an estimated threshold $\Delta E^{\text{thr}}$ for detection. Values were not scaled by the magnitude of the illuminant change, as with some colour constancy indices [47]. Representative values of $\Delta E^{\text{thr}}$ of about 1.0 have been usually quoted for CIELAB space [46,48,49]. But for detecting colour differences in whole images of natural scenes, as distinct from isolated samples, CIELAB threshold values averaged over observers have been estimated [29] as about 2.2, although smaller values have also been obtained [50]. The relevance of these estimates to natural illumination changes is discussed later. The CIELAB threshold was thus set as $\Delta E^{\text{thr}} = 2.2$.

Converting from CIELAB thresholds to CAM02-UCS thresholds is not straightforward since there is no unique mapping between the spaces. But empirically, averaged across the 50 scenes, CIELAB colour differences around 2.2 were found to correspond to CAM02-UCS colour differences of about 1.5. The CAM02-UCS threshold was therefore set as $\Delta E^{\text{thr}} = 1.5$.

## (e) Estimating changes in colour relations

Expected changes in colour relations were characterized by the sizes of the deviations in spatial ratios of cone excitations. These deviations correlate well with observers' performance in discriminating illuminant changes from reflectance changes in an operational approach to colour constancy [51–54].

At each point $(\xi, \eta)$ in the scene, the radiance spectrum $L(\xi, \eta; \lambda)$ was converted into long-, medium-, and short-wavelength-sensitive-cone excitations. Cone spectral sensitivities were taken from the Stockman and Sharpe 2° cone fundamentals [55,56]. For each cone class, let $r_i$ be the ratio of excitations at the $i$th pair of points chosen randomly and independently in the scene.

As with colour differences, this point-based sampling approach is neutral with respect to physical scene structure. Let $r_{i,1}$ and $r_{i,2}$ be the ratios with illuminants 1 and 2 [27]. The observer was not assumed necessarily to be chromatically adapted to each illuminant since ratios of cone excitations are independent of response scaling (within practical limits). To stabilize the variance, the difference $r_{i,1} - r_{i,2}$ was normalized by the mean of $r_{i,1}$ and $r_{i,2}$, giving a relative deviation $\text{RD} = |r_{i,1} - r_{i,2}|/[(r_{i,1} + r_{i,2})/2]$, again averaged over illuminant changes from 4000 K to 6500 K and 25 000 K to 6500 K.

For compatibility with the colour difference data, a threshold RD was derived in the same way as the CAM02-UCS threshold. Averaged across the 50 scenes, CIELAB colour differences around 2.2 corresponded to relative deviations of about 4.8%. The RD threshold was thus set as $\text{RD}^{\text{thr}} = 4.8\%$, which falls between two values estimated in [19].

## (f) Correlation with colorimetric and physical properties

Estimated colour differences and deviations in spatial cone-excitation ratios were tested for an association with the surface colour attributes of chroma and saturation, and, for comparison, the achromatic attribute of lightness, each evaluated under the reference 6500 K illuminant. Chroma measures the colourfulness of an area judged in proportion to the brightness of a similarly illuminated area that appears white, whereas saturation measures the colourfulness of an area judged in proportion to its own brightness (and so is constant with luminance except at very high levels) [41].

These three attributes were quantified by the CIECAM02 model [34], which offers the best descriptor, in particular for saturation with natural scenes [57]. Because deviations in spatial

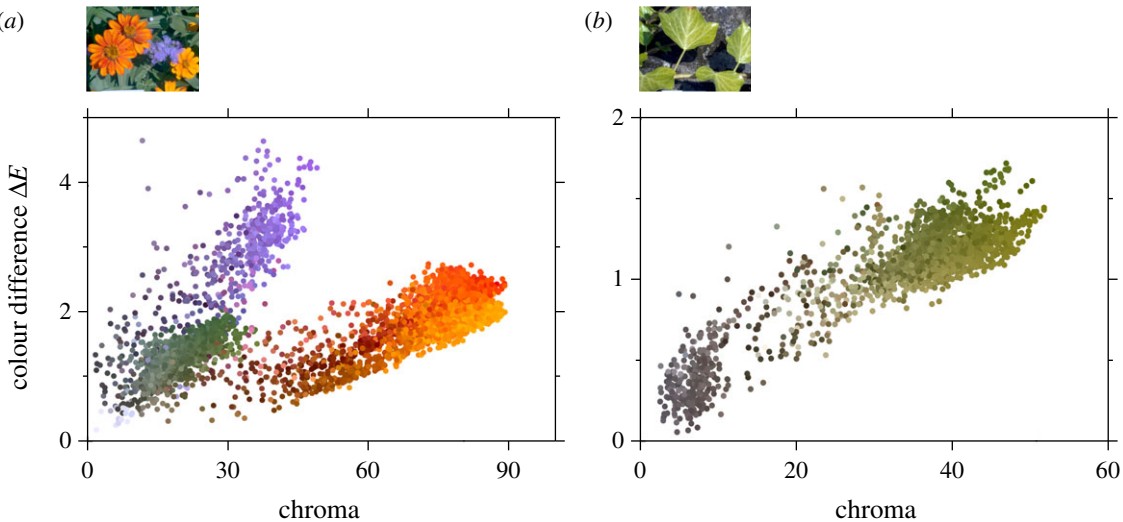

**Figure 3.** Estimated colour differences in two scenes, shown in the thumbnail images. In each panel, CAM02-UCS colour differences $\Delta E$ are plotted against CIECAM02 chroma [34]. Colour differences were averaged over daylight illuminant changes from correlated colour temperatures of 4000 K and 25 000 K to 6500 K. Data points are coloured with the corresponding pixel colours in figure 1. Vertical and horizontal axis scales are adjusted for data ranges. (Online version in colour.)

cone-excitation ratios refer to pairs of points, comparisons were made with the (unsigned) differences in the attribute values at those points.

Estimated colour differences and deviations in spatial cone-excitation ratios were also tested for an association with non-colorimetric physical properties of reflectances.

## (g) Statistics

The main reporting statistics for the 50 scenes are (1) the proportion of scenes in which the upper 5% and upper 10% of expected colour differences or deviations in spatial ratios of cone excitations exceed detection threshold and (2) the median over scenes of the correlations between expected colour differences or deviations in ratios and colorimetric or physical reflectance properties. Correlation was quantified by the Pearson product moment correlation coefficient $\rho$, evaluated where necessary as the square root of the proportion $R^2$ of the variance accounted for in a linear regression. Spearman's rank correlation coefficient offered no advantage. Confidence intervals (CIs) were 95% intervals estimated by Efron's BCa bootstrap method with 1000 bootstrap replications [58].

## 3. Results and comment

## (a) Colour differences in single scenes

Figure 3a shows estimated colour differences for 5000 points drawn randomly from the scene in figure 1. The CAM02-UCS colour difference $\Delta E$, averaged over daylight illuminant changes from CCTs of 4000 K to 6500 K and 25 000 K to 6500 K, is plotted against CIECAM02 chroma for the 6500 K reference illuminant. Data points are coloured with the corresponding pixel colours in figure 1.

About half of the colour differences $\Delta E$ for this scene exceed the CAM02-UCS detection threshold $\Delta E^{\mathrm{thr}} = 1.5$, with the purple flowers producing the largest failures in colour constancy. Values of $\Delta E$ tend to increase with chroma, but at a rate dependent on hue [12]. Because of this dependence, the overall correlation with chroma is reduced, with $\rho = 0.47$.

By contrast, figure 3b shows analogous data for 2000 points drawn randomly from a scene that was more

homogeneous, primarily green foliage (see thumbnail image). Just 1% of the colour differences $\Delta E$ exceed $\Delta E^{\mathrm{thr}} = 1.5$, though they also tend to increase with chroma, and with a much higher correlation, $\rho = 0.85$, than with the more inhomogeneous scene in figure 3a.

These two examples are solely for illustration. In what follows, summary data are presented for the full set of 50 images.

## (b) Frequencies of large colour differences

The locations of the largest CAM02-UCS colour differences $\Delta E$ in each of the 50 scenes are ringed in figure 2. Almost all these values exceeded the estimated detection threshold $\Delta E^{\mathrm{thr}}$, but they give an uncertain guide to the frequency of large colour differences. A more robust measure is provided by percentiles.

Figure 4a,b shows histograms of the upper 5th percentiles of $\Delta E$ in each scene for CAM02-UCS and CIELAB colour differences under the same daylight illuminant changes with a chromatically adapted observer. The vertical dashed lines mark the estimated detection thresholds $\Delta E^{\mathrm{thr}}$.

The proportion of scenes with the upper 5th percentile exceeding threshold was 60% (CI 46% to 72%) with CAM02-UCS and a little higher at 68% (CI 54% to 80%) with CIELAB. Including surfaces with low reflectances as a control (see Methods) changed these proportions by no more than 10%.

As expected, the proportion of scenes with the upper 10th percentile of $\Delta E$ exceeding threshold was smaller, at 44% (CI 30% to 56%) with CAM02-UCS and 54% (CI 40% to 66%) with CIELAB.

If, contrary to assumption, observers were not chromatically adapted, then the proportions of scenes affected increase. Thus, with a default adaptation level [46], the proportion with the upper 10th percentile of $\Delta E$ exceeding threshold was 90% (CI 76% to 96%) with CAM02-UCS and 86% (CI 74% to 94%) with CIELAB. Conversely, if thresholds $\Delta E^{\mathrm{thr}}$ were set larger or changes in illuminant spectrum made smaller, then these proportions decrease.

For clarity, these results do not so much imply that colour constancy does not hold predominantly in these scenes but

*(a)*  *(b)*  *(c)*

**Figure 4.** Histograms of the upper 5th percentiles of (*a*) CAM02-UCS colour differences ΔE [34], (*b*) CIELAB colour differences ΔE [34] and (*c*) relative deviations in spatial ratios of cone excitations RD as a percentage [15] under the same daylight illuminant changes. The vertical dashed lines mark the estimated detection thresholds $\Delta E^{\text{thr}}$ for the three difference measures. Axis scales are adjusted for different bin widths. Data for 50 scenes.

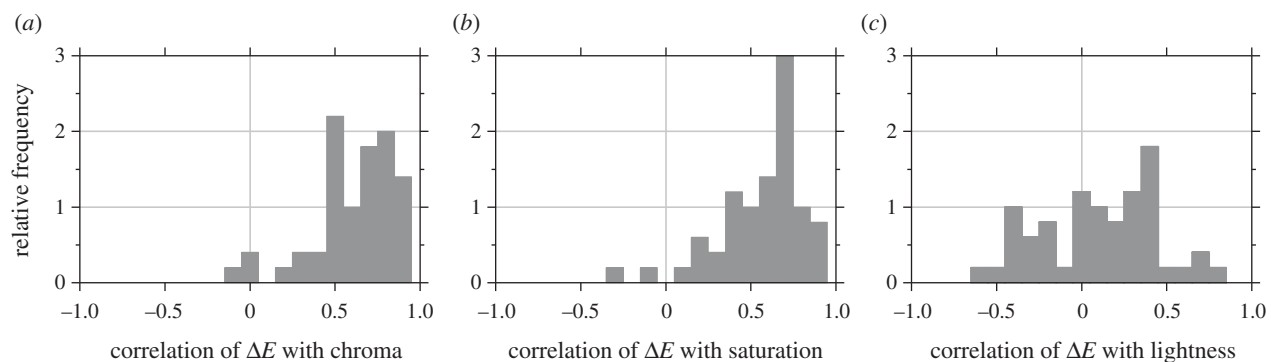

**Figure 5.** Histograms of correlation coefficients between CAM02-UCS colour differences ΔE under daylight illuminant changes and CIECAM02 values of chroma, saturation and lightness [34]. Data for 50 scenes.

that, in around half of them, there exist some individual surfaces or parts of surfaces within those scenes where it is expected to fail.

## (c) Frequencies of large deviations in cone-excitation ratios

Relative deviations (RDs) in spatial cone-excitation ratios were analysed in the same way. Figure 4*c* shows a histogram of the upper 5th percentiles of RDs under the same daylight illuminant changes. The vertical dashed line marks the estimated detection threshold $\Delta E^{\text{thr}}$. The proportion of scenes with the upper 5th percentile exceeding threshold was 80% (CI 68% to 90%), and the proportion with the upper 10th percentile exceeding threshold was 60% (CI 46% to 72%).

Despite the differences between relational colour constancy and colour constancy, the percentiles of RDs and colour differences ΔE were correlated over scenes. For the upper 5th percentiles, $\rho = 0.74$ (CI 0.54 to 0.83) and for the upper 10th percentiles, $\rho = 0.70$ (CI 0.45 to 0.83).

## (d) Colorimetric attributes accounting for variance

Figure 5 shows histograms of the correlation coefficients $\rho$ between CAM02-UCS colour differences ΔE and chroma, saturation and lightness. As noted earlier, appearance attributes were evaluated under the 6500 K illuminant.

For correlations of ΔE with chroma, median $\rho = 0.65$ (CI 0.54 to 0.74), and with saturation, median $\rho = 0.63$ (CI 0.52 to 0.69). For correlations of ΔE with lightness, however, median $\rho = 0.09$ (CI 0.01 to 0.30). The pattern of correlations

was similar for CIELAB colour differences ΔE, with somewhat higher values.

When colour differences were restricted to just lightness differences, so that $\Delta E = \Delta J'$ in CAM02-UCS, the dependence on colourfulness measures persisted. For correlations of $\Delta J'$ with chroma, median $\rho = 0.75$ (CI 0.64 to 0.78) and with saturation, median $\rho = 0.67$ (CI 0.63 to 0.72). For the correlation of $\Delta J'$ with lightness, median $\rho = 0.03$ (CI −0.05 to 0.07). The lack of correlation can be understood with reference to the orange flower in figure 1. The small increase $\Delta J'$ with a change in illuminant is associated not with its lightness but its colourfulness [41].

Although colour differences are associated with colourfulness measures, the plot in figure 3*a* for the flower scene suggests they are also associated with hue. With a linear-circular regression [59] of ΔE on hue angle for each of the 50 scenes, median $\rho = 0.52$ (CI 0.38 to 0.58), lower than with chroma, where median $\rho = 0.65$. Even so, the role of chroma seems not to be secondary to hue angle. In separate linear regressions, the median proportion $R^2$ of variance accounted for by chroma was 43%, by hue angle 27%, and by the two together in a linear and linear-circular regression on chroma and hue angle 66%.

Spatial cone-excitation ratios revealed similar correlations. Figure 6 shows histograms of the correlation coefficients $\rho$ between the relative deviation RD in cone-excitation ratios across a pair of points and the difference in chroma, saturation, and lightness at those points. Illuminant changes were again from 4000 K to 6500 K and 25 000 K to 6500 K and the appearance attributes were evaluated under the 6500 K illuminant.

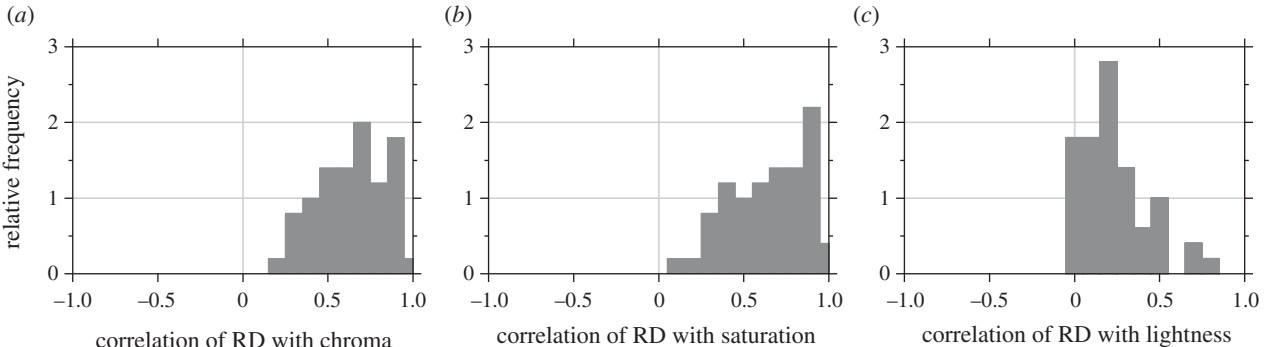

**Figure 6.** Histograms of correlation coefficients between relative deviations RDs in spatial ratios of cone excitations under daylight illuminant changes and differences in CIECAM02 values of chroma, saturation and lightness [34]. Data for 50 scenes.

For chroma, median $\rho = 0.66$ (CI 0.60 to 0.71); for saturation, median $\rho = 0.68$ (CI 0.58 to 0.78); and for lightness, median $\rho = 0.20$ (CI 0.15 to 0.25), paralleling the corresponding medians for correlations of colour differences.

### (e) Reflectance properties accounting for variance

It could be argued that the correlation of estimated colour differences with measures of colourfulness is not due to these attributes per se, but that both are due to the underlying properties of the surface spectral reflectance. Departures from spectral uniformity can be quantified in various ways, for example, by the size of the peaks in the reflectance spectrum, the slopes at each point, and the coefficients of the discrete Fourier transform. How well do any of these properties account for the variance in $\Delta E$ values?

Peak size at each point $(\xi, \eta)$ can be quantified by the average difference $R(\xi, \eta; \lambda_i) - [R(\xi, \eta; \lambda_{i-1}) + R(\xi, \eta; \lambda_{i+1})]/2$ in adjacent reflectance values at each wavelength $\lambda_i$ in the interior of the spectrum. With a linear regression of $\Delta E$ on the maximum peak size in each of the 50 scenes, the median $R^2$ was small, just 5% (CI 1% to 10%).

These low values may be due to broader maxima being discounted, a problem avoided in regressing on slopes rather than peaks. With a linear regression of $\Delta E$ on the maximum absolute slope at each point $(\xi, \eta)$, the median $R^2$ was larger, though still only 10% (CI 6% to 17%), and much less than with chroma, where the median $R^2$ was 43%.

The Fourier coefficients capture instead more extended spectral properties. With a linear regression of $\Delta E$ on the maximum of the Fourier amplitude spectrum at each $(\xi, \eta)$, the median $R^2$ was 44% (CI 35% to 49%), indistinguishable from the value with chroma.

That none of these reflectance properties accounted for more variance than chroma might be because chroma is derived from differences in transformed cone signals. These differences also measure departures from spectral uniformity but moderated by cone spectral sampling characteristics.

## 4. Discussion

Colour constancy failures of some individual surfaces or parts of surfaces seem likely in natural outdoor environments. With a change in the spectrum of a daylight illuminant, 60% of the 50 scenes in figure 2 revealed changes in estimated colour appearance in at least 5% of the surface area and 44% in at least 10% of the surface area, all with a chromatically fully adapted observer. Somewhat higher frequencies of occurrence

were found for changes in colour relations estimated by spatial ratios of cone excitations, independent of the observer's adaptation. Both estimated colour differences and deviations in spatial ratios were correlated with measures of surface colourfulness, and with each other.

Are predicted failures of this kind important in the real-world recognition of objects and surfaces? There are several factors to consider, including the way observers view scenes; the changes in natural illumination; the spatial distributions of surfaces within scenes; and the role of cognitive discounting.

First, no assumptions were made here about an observer's viewing strategy. This was also true for the psychophysical experiments [29,50] used to guide the choice of thresholds for detecting colour differences. In [29], observers judged simply whether a pair of simultaneous side-by-side images of outdoor scenes, one with added colour errors, were the same or different. Similarly, in [50], where images were presented simultaneously and sequentially. Of course, with any individual scene, either outdoors or in the laboratory, detection depends on an observer's colour awareness [60] and on how global colour properties guide attention [53,61,62].

Second, natural illumination changes take many forms. They may be sequential and rapid, as with a cloud passing over the sun or a shift in shadows cast more locally [26,63]; or they can be spatial, with light from different directions [64], or variegated [65] with surfaces partly in direct sunlight and partly in shade [66]. Thresholds for detecting colour differences in these and corresponding laboratory conditions [29,50] depend little on memory [67,68]. By contrast, when illumination changes occur over minutes or hours, as with the spectrum of the direct beam during solar elevation [69], thresholds depend much more on memory and are generally larger [67,68,70,71]. Thresholds are also larger in tasks such as those requiring colour categorization [5].

Third, whether surfaces yielding large colour differences are scattered over the scene or concentrated in one area may affect their detectability. Since localized changes in reflected light can still attract observers' attention by processes that are largely automatic and unconscious [72,73], surface area may not be decisive, as long as it can be resolved. Similar considerations apply to the effect of changes in colour relations [19,51].

Fourth, and last, a case can be made that even if colour differences are detectable, then higher-level, more cognitive processes can discount them. Experimentally, with suitable instructions, observers can indeed separate judgements of colour appearance from judgements of the objective properties of reflecting surfaces under different illuminants [74,75],

though the extent to which the processes are perceptual has been debated [76,77]. Colour constancy may then be construed as the capacity to associate particular colours with particular surface spectral reflectances, in so far as they are defined [35,36]. Crucially, the capacity to make judgements about the origin of perceived changes seems possible only because differences in appearance are attributable to illumination changes that are not coextensive with object surfaces.

This is not the situation here where changes in appearance are localized and bounded by object surfaces. Such changes may be interpreted by observers as unnatural. Thus, in an experiment with chequerboard displays of Munsell surfaces [19], observers compared surfaces undergoing an illuminant change with surfaces undergoing the same illuminant change but where the images were corrected for deviations in spatial cone-excitation ratios. Observers systematically mislabelled the corrected images as appearing more natural, with the probability of mislabelling increasing with the size of the deviations.

There is other observational evidence that localized changes in reflected light, albeit from a different physical process, are perceived as reflectance changes. A tower half lit by a low sun acquires an orange hue ([78], fig. 10), and Ayers Rock, or Uluru, in Australia appears to change 'from bright orange through deep russet to dark purplish-brown' over several minutes at sunset ([79], fig. 9).

One reason that these perceptual effects cannot easily be discounted is that they are optically indistinguishable, or nearly so, from those due to genuine changes in surface reflectance. Given that distinguishing surfaces with different reflectances may be as consequential as recognizing surfaces with the same reflectances, failures of colour constancy with some surfaces may be a necessary trade-off between these two abilities.

Ultimately the limitations of the present approach remain. Despite the widespread use of colour difference metrics in practical applications [41,46], it is possible that colour differences that are obvious in laboratory images are less obvious in the real world. Likewise with deviations in spatial ratios of cone excitations. On the other hand, as indicated earlier, real-world changes in the spatial distribution of light [27] may actually be more disruptive than purely spectral changes.

Notwithstanding these uncertainties, failures of colour constancy in outdoor environments should be expected with some individual surfaces or parts of surfaces, especially colourful ones. Rather than representing a fundamental lapse of veridical perception, however, these failures may be a by-product of sensory mechanisms or processes designed to detect genuine changes in surface reflectance, whenever they occur.

Data accessibility. The hyperspectral reflectance data used in this study are available at https://doi.org/10.48420/14877285 [80].

Authors' contributions. D.H.F.: data curation, formal analysis, software, writing—original draft; A.R. conceptualization, writing—original draft, writing—review and editing.

All authors gave final approval for publication and agreed to be held accountable for the work performed therein.

Competing interests. We declare we have no competing interests.

Funding. Part of this work was supported by the Engineering and Physical Sciences Research Council (GR/R39412/01, EP/B000257/1).

Acknowledgements. We thank V. Cheung, M. R. Pointer, and S. Westland for advice; K .R. Gegenfurtner and C. Witzel for additional data on constancy indices; K. Amano and S. M. C. Nascimento for collaborating in the acquisition and processing of the hyperspectral images; and K. Amano and K. R. Gegenfurtner for critically reading the manuscript. A partial report of these findings was presented at the 25th Symposium of the International Colour Vision Society, Riga, Latvia, 2019.

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
