## [Peer Review File · Proceedings of the Royal Society B: Biological Sciences]

Review History

RSPB-2021-1481.R0 (Original submission)

Review form: Reviewer 1

Recommendation

Major revision is needed (please make suggestions in comments)

Scientific importance: Is the manuscript an original and important contribution to its field?

Excellent

General interest: Is the paper of sufficient general interest?

Excellent

Quality of the paper: Is the overall quality of the paper suitable?

Good

Is the length of the paper justified?

Yes

Should the paper be seen by a specialist statistical reviewer?

No

Do you have any concerns about statistical analyses in this paper? If so, please specify them explicitly in your report.

No

It is a condition of publication that authors make their supporting data, code and materials available - either as supplementary material or hosted in an external repository. Please rate, if applicable, the supporting data on the following criteria.

Is it accessible?

No

Is it clear?

N/A

Is it adequate?

N/A

Do you have any ethical concerns with this paper?

No

Comments to the Author

The authors tested “whether failures in colour constancy are common in outdoor environments and whether they are related to the colourfulness of surfaces”. They performed their test through computational modeling. They rendered hyperspectral images under different simulated illuminations and used a state-of-the-art color appearance model to predict color appearance for each pixel. In the majority of images (88%) there was at least one “part of a surface” that changed appearance.

In my view, this is a highly interesting and rigorous modeling exercise that illustrates well the problems when dealing with color constancy in the natural world. I have several specific comments below, but mainly I see a few conceptual issues that the authors could solve by more explicitly stating their assumptions, and considering alternative interpretations.

(1) My main issue is that the authors make statements about color constancy in outdoor scenes, without ever making any measurement with a human observer. They replace the human observer with two color appearance models, CIELAB and CIECAM02. While these are the best models currently available, they should not be considered as 100% equivalents of human observers when it comes to detailed analyses of color differences in natural scenes. For example, while CIECAM02 performed best to predict human saturation judgments in natural scenes (ref 54), it was by no means perfect in doing so. The authors right now assume that CIECAM02 is perfect and that failures of color constancy are abundant. Someone else might assume that human color constancy in natural scenes is perfect and that the CIECAM02 model fails for such scenes. It could be discussed which is more likely, and I think the paper would benefit enormously from such a discussion.

(2) A second conceptual issue concerns color appearance versus color constancy. It could be argued that color constancy allows us to assign colors to *objects*, where all pixels within an object have the same spectral reflectance. Of course, we can still see chromatic differences within an object that are caused by shading, interreflections or translucency. In the example image of the red and yellow roses (row 2, column 2 in Figure 2), interreflections cause dark and highly saturated regions. In the image with the leaves (row 4, column 1 in Figure 2), translucency and shading causes large variations in luminance. Still, we would assign the same unique color to the leaves and the roses. In the analysis of the hyperspectral images, the pixels within an object are assigned different spectral reflectances, so their simulated color coordinates under a different illuminant are not necessarily exactly identical to the color coordinates that would be obtained if the same three-dimensional scene would be illuminated differently. I admit that it is unclear how

large these deviations would be. The authors could investigate the issue by checking the magnitude of deviations from a constant color appearance pixelwise in the images. For example, are the deviations largest in regions prone to interreflections?

(3) A third problematic point concerns the values for the ΔE thresholds. Low values are typically used when the visibility of abutting colored surfaces is judged, i.e. when color sensitivity is largest. When a spatial or temporal distance is involved, values are much larger and estimates in the literature are rather divergent. Even a ΔE of 4.7 might not lead to a noticeable failure of color constancy under such conditions (see Figure 3), while the two surfaces are clearly different when viewed simultaneously and neutrally adapted (see blue-purple flower in Figure 1). The authors state this need for larger values when memory is involved, but they never apply such higher values. In my view, Figure 4 shows that the 95th percentile of ΔE mainly is below a reasonable threshold, indicating that color constancy would predominantly hold in natural scenes.

(4) I think the focus on the pixel with the maximum ΔE is not adequate, given the above constraints with the hyperspectral images. The 95th percentile, or even the 90th percentile, would suffice.

Minor comments:

Introduction: 88% of scenes. is that the lower bound of the CI on top of page 10? How was the CI computed?

Methods: "spatial coordinates u, v (not to be confused with chromatic coordinates)."

Why not use other letters then for the spatial coordinates where the confusion could not arise (i, j) for example.

Figure 2: You could mark the images and image regions with the biggest ΔE values.

Review form: Reviewer 2 (Bevil Conway)

Recommendation

Major revision is needed (please make suggestions in comments)

Scientific importance: Is the manuscript an original and important contribution to its field?

Acceptable

General interest: Is the paper of sufficient general interest?

Acceptable

Quality of the paper: Is the overall quality of the paper suitable?

Acceptable

Is the length of the paper justified?

Yes

Should the paper be seen by a specialist statistical reviewer?

No

Do you have any concerns about statistical analyses in this paper? If so, please specify them explicitly in your report.

No

It is a condition of publication that authors make their supporting data, code and materials available - either as supplementary material or hosted in an external repository. Please rate, if applicable, the supporting data on the following criteria.

Is it accessible?

No

Is it clear?

No

Is it adequate?

No

Do you have any ethical concerns with this paper?

No

Comments to the Author

This paper describes how colors of natural surfaces might be expected to change with different lighting, using a computational approach in which colors of pixels in hyperspectral images of natural scenes are evaluated using a color-appearance model. The research is technically sound, and the results fill a gap in knowledge. Importantly, it is valuable to know to what extent possible failures in color constancy occur (especially in natural scenes), and what such observations suggest or imply about the limits of color behavior (such as color memory). Four major comments:

First, failures in matching colors across simulated lighting conditions could either point to failures of color constancy or failures of the color-appearance model. It isn't clear that the results can distinguish these possibilities. It's possible that the analysis of cone ratios helps resolve this quandary (ref. 22), but more discussion would help. Regardless, it would be important to provide a caveat regarding the conclusion about the failures of color constancy. The failures described are those one might expect *if* color appearance models are accurate. Given documented failures in color-appearance models (and the unreliability of estimating color appearance relationships over large distances in any color space), there is a decent argument that this assumption is not supported.

Second, one of the conclusions is that failures of color constancy in natural scenes are common, and that they are correlated with saturation. To what extent is this result a consequence of the dynamic range of the system: as colors become more saturated, there is more cone contrast in the stimulus and therefore more scope for them to change in color under different illuminants. Put another way, a peaky reflectance function will be more likely to appear a different color under different spectral distributions of illumination than a broad-spectrum reflectance function. The authors write "when constancy failures do occur, are they attributable to surface colourfulness". One is not because of the other, but rather both are attributed to spectral peakiness. Related, it's been shown that failures of constancy in unnatural scenes correlate with saturation (as predicted), so a key question that doesn't seem to be clearly addressed in the present manuscript is why one might expect this relationship not to hold for natural images. These considerations open up a set of unanswered questions: to what extent is the paper about quantifying the relative saturation of surfaces in natural scenes? Is this necessarily any different for natural scenes than artificial scenes? Are the natural scenes representative, or instead biased by decisions made by humans (the images are of only one geographic location and there appears to be an enrichment of salient objects)?

Third, methodologically, the paper assumes a single, uniform illuminant. The computational approach with these simplifying assumptions is useful, but is the rationale for these assumptions sound? As the authors point out in their introduction, measurements of color constancy should be made under the conditions in which the surfaces/scenes are normally viewed, which is why

the gap in estimates of color constancy for natural scenes is glaring. But this logic means the measurements should correspond to the other conditions of the lighting, which in natural scenes involves dynamic changes (of not unlimited variety) and spatial variations (along the daylight locus). The results are valuable because of the potential relevance to understanding behavior, and if behavior is incapable of discerning changes in color under the natural dynamic conditions of natural scenes, then it's unclear whether the lapses of the color-appearance model constitute "failures" of constancy.

Fourth, the paper alludes to an observation that hue predicts some of the extent of the failure in color constancy. I think this is potentially very interesting and deserving of a figure, especially in light of other work relating to the color statistics of natural scenes and objects.

Other.

Pg. 7. Some more information on the sources of the "noise" would be helpful.

Are the hyperspectral images and other materials available in a public repository? I tried to open the link and it appeared to be private or broken.

Figure 2 is nicely illustrative, and I wonder if one could be possible to present the full data in a somewhat analogous way. For example, why can't we have a scatter plot of the form as Figure 2 but for all images, e.g. randomly draw 10,000 pixels from across the 50 images, so we have a nice selection of colors, do it a bunch of times (i.e. different draws of 10,000 pixels), and you can see from the panels the relationship of the color-appearance shift with hue (and chroma)?

Decision letter (RSPB-2021-1481.R0)

10-Aug-2021

Dear Professor Foster:

I am writing to inform you that your manuscript RSPB-2021-1481 entitled "Colourful failures of environmental colour constancy" has, in its current form, been rejected for publication in Proceedings B.

This action has been taken on the advice of referees, who have recommended that substantial revisions are necessary. With this in mind we would be happy to consider a resubmission, provided the comments of the referees are fully addressed. However please note that this is not a provisional acceptance.

- 1) A 'response to referees' document including details of how you have responded to the comments, and the adjustments you have made.
- 2) A clean copy of the manuscript and one with 'tracked changes' indicating your 'response to referees' comments document.

- 3) Line numbers in your main document.
- 4) Data - please see our policies on data sharing to ensure that you are complying (<https://royalsociety.org/journals/authors/author-guidelines/#data>).

Sincerely,
 Dr Robert Barton
 mailto: proceedingsb@royalsociety.org

Associate Editor
 Board Member: 1
 Comments to Author:

We have now heard from the two experts who reviewed your manuscript. Both had quite positive things to say about the manuscript, but they both raised serious conceptual and methodological issues that need to be addressed. They were also unable to access the data through the link provided. Was the site private and/or protected rather than public? It would also be good if the code used for the analysis were made available.

Reviewer(s)' Comments to Author:

Referee: 1

Comments to the Author(s)

The authors tested "whether failures in colour constancy are common in outdoor environments and whether they are related to the colourfulness of surfaces". They performed their test through computational modeling. They rendered hyperspectral images under different simulated illuminations and used a state-of-the-art color appearance model to predict color appearance for each pixel. In the majority of images (88%) there was at least one "part of a surface" that changed appearance.

In my view, this is a highly interesting and rigorous modeling exercise that illustrates well the problems when dealing with color constancy in the natural world. I have several specific comments below, but mainly I see a few conceptual issues that the authors could solve by more explicitly stating their assumptions, and considering alternative interpretations.

(1) My main issue is that the authors make statements about color constancy in outdoor scenes, without ever making any measurement with a human observer. They replace the human observer with two color appearance models, CIELAB and CIECAM02. While these are the best models currently available, they should not be considered as 100% equivalents of human observers when it comes to detailed analyses of color differences in natural scenes. For example, while CIECAM02 performed best to predict human saturation judgments in natural scenes (ref 54), it was by no means perfect in doing so. The authors right now assume that CIECAM02 is perfect and that failures of color constancy are abundant. Someone else might assume that human color constancy in natural scenes is perfect and that the CIECAM02 model fails for such scenes. It could be discussed which is more likely, and I think the paper would benefit enormously from such a discussion.

(2) A second conceptual issue concerns color appearance versus color constancy. It could be argued that color constancy allows us to assign colors to *objects*, where all pixels within an object have the same spectral reflectance. Of course, we can still see chromatic differences within an object that are caused by shading, interreflections or translucency. In the example image of the red and yellow roses (row 2, column 2 in Figure 2), interreflections cause dark and highly

saturated regions. In the image with the leaves (row 4, column 1 in Figure 2), translucency and shading causes large variations in luminance. Still, we would assign the same unique color to the leaves and the roses. In the analysis of the hyperspectral images, the pixels within an object are assigned different spectral reflectances, so their simulated color coordinates under a different illuminant are not necessarily exactly identical to the color coordinates that would be obtained if the same three-dimensional scene would be illuminated differently. I admit that it is unclear how large these deviations would be. The authors could investigate the issue by checking the magnitude of deviations from a constant color appearance pixelwise in the images. For example, are the deviations largest in regions prone to interreflections?

(3) A third problematic point concerns the values for the ΔE thresholds. Low values are typically used when the visibility of abutting colored surfaces is judged, i.e. when color sensitivity is largest. When a spatial or temporal distance is involved, values are much larger and estimates in the literature are rather divergent. Even a ΔE of 4.7 might not lead to a noticeable failure of color constancy under such conditions (see Figure 3), while the two surfaces are clearly different when viewed simultaneously and neutrally adapted (see blue-purple flower in Figure 1). The authors state this need for larger values when memory is involved, but they never apply such higher values. In my view, Figure 4 shows that the 95th percentile of ΔE mainly is below a reasonable threshold, indicating that color constancy would predominantly hold in natural scenes.

(4) I think the focus on the pixel with the maximum ΔE is not adequate, given the above constraints with the hyperspectral images. The 95th percentile, or even the 90th percentile, would suffice.

Minor comments:

Introduction: 88% of scenes. is that the lower bound of the CI on top of page 10? How was the CI computed?

Methods: "spatial coordinates u, v (not to be confused with chromatic coordinates)."

Why not use other letters than for the spatial coordinates where the confusion could not arise (i, j) for example.

Figure 2: You could mark the images and image regions with the biggest ΔE values.

Referee: 2

Comments to the Author(s)

This paper describes how colors of natural surfaces might be expected to change with different lighting, using a computational approach in which colors of pixels in hyperspectral images of natural scenes are evaluated using a color-appearance model. The research is technically sound, and the results fill a gap in knowledge. Importantly, it is valuable to know to what extent possible failures in color constancy occur (especially in natural scenes), and what such observations suggest or imply about the limits of color behavior (such as color memory). Four major comments:

First, failures in matching colors across simulated lighting conditions could either point to failures of color constancy or failures of the color-appearance model. It isn't clear that the results can distinguish these possibilities. It's possible that the analysis of cone ratios helps resolve this quandary (ref. 22), but more discussion would help. Regardless, it would be important to provide a caveat regarding the conclusion about the failures of color constancy. The failures described are those one might expect *if* color appearance models are accurate. Given documented failures in color-appearance models (and the unreliability of estimating color appearance relationships over large distances in any color space), there is a decent argument that this assumption is not supported.

Second, one of the conclusions is that failures of color constancy in natural scenes are common, and that they are correlated with saturation. To what extent is this result a consequence of the dynamic range of the system: as colors become more saturated, there is more cone contrast in the stimulus and therefore more scope for them to change in color under different illuminants. Put another way, a peaky reflectance function will be more likely to appear a different color under different spectral distributions of illumination than a broad-spectrum reflectance function. The authors write “when constancy failures do occur, are they attributable to surface colourfulness”. One is not because of the other, but rather both are attributed to spectral peakiness. Related, it’s been shown that failures of constancy in unnatural scenes correlate with saturation (as predicted), so a key question that doesn’t seem to be clearly addressed in the present manuscript is why one might expect this relationship not to hold for natural images. These considerations open up a set of unanswered questions: to what extent is the paper about quantifying the relative saturation of surfaces in natural scenes? Is this necessarily any different for natural scenes than artificial scenes? Are the natural scenes representative, or instead biased by decisions made by humans (the images are of only one geographic location and there appears to be an enrichment of salient objects)?

Third, methodologically, the paper assumes a single, uniform illuminant. The computational approach with these simplifying assumptions is useful, but is the rationale for these assumptions sound? As the authors point out in their introduction, measurements of color constancy should be made under the conditions in which the surfaces/scenes are normally viewed, which is why the gap in estimates of color constancy for natural scenes is glaring. But this logic means the measurements should correspond to the other conditions of the lighting, which in natural scenes involves dynamic changes (of not unlimited variety) and spatial variations (along the daylight locus). The results are valuable because of the potential relevance to understanding behavior, and if behavior is incapable of discerning changes in color under the natural dynamic conditions of natural scenes, then it’s unclear whether the lapses of the color-appearance model constitute “failures” of constancy.

Fourth, the paper alludes to an observation that hue predicts some of the extent of the failure in color constancy. I think this is potentially very interesting and deserving of a figure, especially in light of other work relating to the color statistics of natural scenes and objects.

Other.

Pg. 7. Some more information on the sources of the “noise” would be helpful.

Are the hyperspectral images and other materials available in a public repository? I tried to open the link and it appeared to be private or broken.

Figure 2 is nicely illustrative, and I wonder if one could be possible to present the full data in a somewhat analogous way. For example, why can't we have a scatter plot of the form as Figure 2 but for all images, e.g. randomly draw 10,000 pixels from across the 50 images, so we have a nice selection of colors, do it a bunch of times (i.e. different draws of 10,000 pixels), and you can see from the panels the relationship of the color-appearance shift with hue (and chroma)?

Author's Response to Decision Letter for (RSPB-2021-1481.R0)

See Appendix A.

RSPB-2021-2483.R0

Review form: Reviewer 1

Recommendation

Accept with minor revision (please list in comments)

Scientific importance: Is the manuscript an original and important contribution to its field?

Excellent

General interest: Is the paper of sufficient general interest?

Excellent

Quality of the paper: Is the overall quality of the paper suitable?

Excellent

Is the length of the paper justified?

Yes

Should the paper be seen by a specialist statistical reviewer?

No

Do you have any concerns about statistical analyses in this paper? If so, please specify them explicitly in your report.

No

It is a condition of publication that authors make their supporting data, code and materials available - either as supplementary material or hosted in an external repository. Please rate, if applicable, the supporting data on the following criteria.

Is it accessible?

Yes

Is it clear?

Yes

Is it adequate?

No

Do you have any ethical concerns with this paper?

No

Comments to the Author

The authors did an excellent job in revising the paper. But while it is all technically correct, I do struggle with the conclusion. If color constancy should fail in the real world, it will ALWAYS include a memory component. We never see two visual worlds side by side. Thus, a much higher delta E threshold would really be more appropriate, maybe twice as high as the ones used now. I would suggest the authors include such a higher value in their Figure 4. I suspect the conclusion might then be that failures are exceedingly rare.

Some of the other points I made in my earlier review are dealt with now, but the answers seem to be well hidden. For example, how much of the predicted deviations could be due to the color appearance models used? This is a question many readers may ask.

Review form: Reviewer 2

Recommendation

Accept as is

Scientific importance: Is the manuscript an original and important contribution to its field?

Good

General interest: Is the paper of sufficient general interest?

Good

Quality of the paper: Is the overall quality of the paper suitable?

Excellent

Is the length of the paper justified?

Yes

Should the paper be seen by a specialist statistical reviewer?

No

Do you have any concerns about statistical analyses in this paper? If so, please specify them explicitly in your report.

No

It is a condition of publication that authors make their supporting data, code and materials available - either as supplementary material or hosted in an external repository. Please rate, if applicable, the supporting data on the following criteria.

Is it accessible?

Yes

Is it clear?

Yes

Is it adequate?

Yes

Do you have any ethical concerns with this paper?

No

Comments to the Author

The authors have done a good job addressing the comments from me and the other reviewer, and the article makes a useful contribution to understanding likely failures of color constancy under natural conditions.

Decision letter (RSPB-2021-2483.R0)

15-Dec-2021

Dear Professor Foster:

Your manuscript has now been peer reviewed and the reviews have been assessed by an Associate Editor. The reviewers' comments (not including confidential comments to the Editor)

and the comments from the Associate Editor are included at the end of this email for your reference. As you will see, the reviewers and the Editors have raised some concerns with your manuscript and we would like to invite you to revise your manuscript to address them.

Research ethics:

Use of animals and field studies:

It is a condition of publication that you make available the data and research materials supporting the results in the article (<https://royalsociety.org/journals/authors/author-guidelines/#data>). Datasets should be deposited in an appropriate publicly available repository and details of the associated accession number, link or DOI to the datasets must be included in the Data Accessibility section of the article (<https://royalsociety.org/journals/ethics-policies/data-sharing-mining/>). Reference(s) to datasets should also be included in the reference list of the article with DOIs (where available).

Please submit a copy of your revised paper within three weeks. If we do not hear from you within this time your manuscript will be rejected. If you are unable to meet this deadline please let us know as soon as possible, as we may be able to grant a short extension.

Best wishes,
Dr Robert Barton
<mailto:proceedingsb@royalsociety.org>

Associate Editor
Comments to Author:

We have now heard back from the two experts who reviewed your re-submitted manuscript. I am pleased to say they are both positive. Reviewer 1 has some remaining concerns that you will have to deal with before we can move forward, however.

Reviewer(s)' Comments to Author:

Referee: 2

Comments to the Author(s).

The authors have done a good job addressing the comments from me and the other reviewer, and the article makes a useful contribution to understanding likely failures of color constancy under natural conditions.

Referee: 1

Comments to the Author(s).

The authors did an excellent job in revising the paper. But while it is all technically correct, I do struggle with the conclusion. If color constancy should fail in the real world, it will ALWAYS include a memory component. We never see two visual worlds side by side. Thus, a much higher delta E threshold would really be more appropriate, maybe twice as high as the ones used now. I would suggest the authors include such a higher value in their Figure 4. I suspect the conclusion might then be that failures are exceedingly rare.

Some of the other points I made in my earlier review are dealt with now, but the answers seem to be well hidden. For example, how much of the predicted deviations could be due to the color appearance models used? This is a question many readers may ask.

Author's Response to Decision Letter for (RSPB-2021-2483.R0)

See Appendix B.

Decision letter (RSPB-2021-2483.R1)

04-Jan-2022

Dear Professor Foster

I am pleased to inform you that your manuscript entitled "Colour constancy failures expected in colourful environments" has been accepted for publication in Proceedings B.

Data Accessibility section

Open Access

Paper charges

All supplementary materials accompanying an accepted article will be treated as in their final form. They will be published alongside the paper on the journal website and posted on the online

figshare repository. Files on figshare will be made available approximately one week before the accompanying article so that the supplementary material can be attributed a unique DOI.

Sincerely,
Dr Robert Barton
Editor, Proceedings B
mailto: proceedingsb@royalsociety.org

Associate Editor:

Board Member

Comments to Author:

Thank you for your revised manuscript. You have done an admirable job in dealing with the remaining concerns.

Appendix A

RESPONSE TO REFEREES

Our responses are in blue. Points are answered in order of occurrence.

Associate Editor

Board Member: 1

Comments to Author:

We have now heard from the two experts who reviewed your manuscript. Both had quite positive things to say about the manuscript, but they both raised serious conceptual and methodological issues that need to be addressed. They were also unable to access the data through the link provided. Was the site private and/or protected rather than public? It would also be good if the code used for the analysis were made available.

We are grateful for the reviewers' comments. Our responses are interleaved.

As for data access, the original upload was apparently incomplete but not reported by figshare. The upload has now been repeated and tested. The link (ms line 475) remains private but should be accessible to the reviewers. If the submission is accepted, it will be made public.

Reviewer(s)' Comments to Author:

Referee: 1

Comments to the Author(s)

The authors tested “whether failures in colour constancy are common in outdoor environments and whether they are related to the colourfulness of surfaces”. They performed their test through computational modeling. They rendered hyperspectral images under different simulated illuminations and used a state-of-the-art color appearance model to predict color appearance for each pixel. In the majority of images (88%) there was at least one “part of a surface” that changed appearance.

In my view, this is a highly interesting and rigorous modeling exercise that illustrates well the problems when dealing with color constancy in the natural world. I have several specific comments below, but mainly I see a few conceptual issues that the authors could solve by more explicitly stating their assumptions, and considering alternative interpretations.

We have now clarified our assumptions at each stage. We start with colour differences and introduce appearance attributes (saturation, chroma, etc.) as explanatory variables only when needed. The reviewer's concerns still apply, but we believe this account is simpler.

(1) My main issue is that the authors make statements about color constancy in outdoor scenes, without ever making any measurement with a human observer.

We now say more about why human experiments outdoors are difficult and why modelling is necessary (lines 89-93 in the revised ms). We have also made changes to the text throughout to indicate that our estimates are predictions.

They replace the human observer with two color appearance models, CIELAB and CIECAM02. While these are the best models currently available, they should not be considered as 100% equivalents of human observers when it comes to detailed analyses of color differences in natural scenes. For example, while CIECAM02 performed best to predict human saturation judgments in natural scenes (ref 54), it was by no means perfect in doing so. The authors right now assume that CIECAM02 is perfect and that failures of color constancy are abundant. Someone else might assume that human color constancy in natural scenes is perfect and that the CIECAM02 model fails for such scenes. It could be discussed which is more likely, and I think the paper would benefit enormously from such a discussion.

We have now introduced material in the Introduction (lines 95-96) and Methods (lines 159-162) explaining our use of colour difference metrics as a measure rather than as a model. We also identify the question of extrapolation from the laboratory to the real world as a potential limitation in the Discussion (lines 440-442).

(2) A second conceptual issue concerns color appearance versus color constancy. It could be argued that color constancy allows us to assign colors to *objects*, where all pixels within an object have the same spectral reflectance. Of course, we can still see chromatic differences within an object that are caused by shading, interreflections or translucency. In the example image of the red and yellow roses (row 2, column 2 in Figure 2), interreflections cause dark and highly saturated regions. In the image with the leaves (row 4, column 1 in Figure 2), translucency and shading causes large variations in luminance. Still, we would assign the same unique color to the leaves and the roses.

We sympathize with this object-oriented interpretation of colour constancy, but what we are trying to show is that distinguishing between an illumination change and a reflectance change can be difficult when the perceptual effects are coextensive. We have now mentioned this object-oriented interpretation of colour constancy and expanded our argument (lines 413-422).

In the analysis of the hyperspectral images, the pixels within an object are assigned different spectral reflectances, so their simulated color coordinates under a different illuminant are not necessarily exactly identical to the color coordinates that would be obtained if the same three-dimensional scene would be illuminated differently. I admit that it is unclear how large these deviations would be. The authors could investigate the issue by checking the magnitude of deviations from a constant color appearance pixelwise in the images. For example, are the deviations largest in regions prone to interreflections?

We have made it clearer that applying the same simulated change in illumination spectrum at each point in the scene avoids the spatial confounds associated with real illumination change (lines 122-131, and supplementary material).

We have also marked the locations of the largest colour errors in a revised Fig. 2. These points seem not to be particularly associated with interreflections.

(3) A third problematic point concerns the values for the ΔE thresholds. Low values are typically used when the visibility of abutting colored surfaces is judged, i.e. when color sensitivity is largest. When a spatial or temporal distance is involved, values are much larger and estimates in the literature are rather divergent. Even a ΔE of 4.7 might not lead to a noticeable failure of color constancy under such conditions (see Figure 3), while the two surfaces are clearly different when viewed simultaneously and neutrally adapted (see blue-purple flower in Figure 1).

We have now drawn attention to the fact that colour-difference measures and values for ΔE were based on laboratory experiments in which observers compared whole scenes presented simultaneously or sequentially, rather than individual abutting surfaces (lines 159-162).

The authors state this need for larger values when memory is involved, but they never apply such higher values.

We have moved this contextual note to the Discussion (lines 402-404).

In my view, Figure 4 shows that the 95th percentile of ΔE mainly is below a reasonable threshold, indicating that color constancy would predominantly hold in natural scenes.

I think we originally said in the text that “95th percentiles of ΔE ... exceeded ΔE thr in 46% of scenes”. Our point, which we have made clearer (lines 295-297), is that our results do not imply that colour constancy does not predominantly hold in natural scenes, but that in around half of them, it is possible to find surfaces or parts of surfaces where it fails.

To verify the percentile estimates, we recalculated the CIECAM02 threshold value and reran the simulations. Because the CAM02-UCS differences distribution peaks around the threshold, the proportion either side is potentially unstable, evidenced by the large confidence intervals (lines 288-294).

(4) I think the focus on the pixel with the maximum ΔE is not adequate, given the above constraints with the hyperspectral images. The 95th percentile, or even the 90th percentile, would suffice.

We have removed the histogram of maximum ΔE values and show only the histogram of 95th percentiles (Fig. 4) and report numerical results for both 95th and 90th percentiles (e.g. lines 288-294).

Minor comments:

Introduction: 88% of scenes. is that the lower bound of the CI on top of page 10? How was the CI computed?

The 88% was a transcription error. It has been removed as we summarize results only for the 95th and 90th percentiles of ΔE values (lines 27-289).

Computation of the bootstrap CIs is explained in the Methods (lines 249-250).

Methods: “spatial coordinates u,v (not to be confused with chromatic coordinates).”
Why not use other letters then for the spatial coordinates where the confusion could not arise (i, j) for example.

Most letter pairs from the Latin alphabet have special associations, so we have replaced u,v by the Greek ξ, η .

Figure 2: You could mark the images and image regions with the biggest ΔE values.

We have put white circles around points in Fig. 2 where ΔE is maximum.

Referee: 2

Comments to the Author(s)

This paper describes how colors of natural surfaces might be expected to change with different lighting, using a computational approach in which colors of pixels in hyperspectral images of natural scenes are evaluated using a color-appearance model. The research is technically sound, and the results fill a gap in knowledge. Importantly, it is valuable to know to what extent possible failures in color constancy occur (especially in natural scenes), and what such observations suggest or imply about the limits of color behavior (such as color memory). Four major comments:

First, failures in matching colors across simulated lighting conditions could either point to failures of color constancy or failures of the color-appearance model. It isn't clear that the results can distinguish these possibilities. It's possible that the analysis of cone ratios helps resolve this quandary (ref. 22), but more discussion would help.

We have now introduced material in the Introduction (lines 95-96) and Methods (lines 159-162) explaining our use of colour difference metrics as a measure rather than as a model.

Regardless, it would be important to provide a caveat regarding the conclusion about the failures of color constancy. The failures described are those one might expect *if* color appearance models are accurate. Given documented failures in color-appearance models (and the unreliability of estimating color appearance relationships over large distances in any color space), there is a decent argument that this assumption is not supported.

We also identify the question of extrapolation from laboratory to the real world as a potential limitation in the Discussion, along with other uncertainties (lines 440-446).

Second, one of the conclusions is that failures of color constancy in natural scenes are common, and that they are correlated with saturation. To what extent is this result a consequence of the dynamic range of the system: as colors become more saturated, there is

more cone contrast in the stimulus and therefore more scope for them to change in color under different illuminants. Put another way, a peaky reflectance function will be more likely to appear a different color under different spectral distributions of illumination than a broad-spectrum reflectance function.

We have added a new section 3 (e) where we analyse the fits of different measures of spectral nonuniformity (lines 361-381).

The authors write “when constancy failures do occur, are they attributable to surface colourfulness”. One is not because of the other, but rather both are attributed to spectral peakiness.

Nonuniformity is indeed a common factor, but for the measures we tried, they do no better than chroma, for potentially good reasons (lines 378-381). But we do now refer to failures correlating or being associated with measures of colourfulness rather than being attributed (e.g. lines 389-390).

Related, it’s been shown that failures of constancy in unnatural scenes correlate with saturation (as predicted), so a key question that doesn’t seem to be clearly addressed in the present manuscript is why one might expect this relationship not to hold for natural images.

The questions we wanted to address was whether constancy failures occur relatively often in natural scenes (they might be extremely rare) and if they do whether colourfulness is still a good explanatory factor. We have sharpened the phrasing (lines 84-88) and have pointed to the fact that previous reports have been based on Munsell reflectances and idealized spectra, whose gamuts differ from those of the real world.

These considerations open up a set of unanswered questions: to what extent is the paper about quantifying the relative saturation of surfaces in natural scenes? Is this necessarily any different for natural scenes than artificial scenes?

See preceding response.

Are the natural scenes representative, or instead biased by decisions made by humans (the images are of only one geographic location and there appears to be an enrichment of salient objects)?

They were recorded in several locations in Portugal but as we point out in the revision they include major land-cover classes (lines 101-107). As for an enrichment of salient objects, it could be argued that any scene that isn’t uniform will contain salient objects. In fact constancy errors seem also to occur at nonsalient points (see revised Fig. 2). We have added a comment about salience (lines 443-445).

Third, methodologically, the paper assumes a single, uniform illuminant. The computational approach with these simplifying assumptions is useful, but is the rationale for these assumptions sound? As the authors point out in their introduction, measurements of color

constancy should be made under the conditions in which the surfaces/scenes are normally viewed, which is why the gap in estimates of color constancy for natural scenes is glaring.

Our underlying assumption is that the spectral change in illuminant should be the same at each point in the scene. Rather than just citing an earlier analysis, we now point out (lines 122-131) that multiplying an effective spectral reflectance by successive uniform illuminants is equivalent to multiplying the true local spectral reflectance by true successive local illuminants, providing that the spectral change is the same for each. Mathematical arguments are given in the supplementary material.

Our response to the next question is also relevant here.

But this logic means the measurements should correspond to the other conditions of the lighting, which in natural scenes involves dynamic changes (of not unlimited variety) and spatial variations (along the daylight locus). The results are valuable because of the potential relevance to understanding behavior, and if behavior is incapable of discerning changes in color under the natural dynamic conditions of natural scenes, then it's unclear whether the lapses of the color-appearance model constitute "failures" of constancy.

This bears on the question of cognitive discounting, which we mention in the revised version (lines 413-422). It is an empirical issue that we cannot resolve here. Our concern in this analysis is solely with how surface reflectances limit constancy, not with the additional effects of changing illumination geometry.

Fourth, the paper alludes to an observation that hue predicts some of the extent of the failure in color constancy. I think this is potentially very interesting and deserving of a figure, especially in light of other work relating to the color statistics of natural scenes and objects.

See reply to last point.

Other.

Pg. 7. Some more information on the sources of the "noise" would be helpful.

We now mention non-imaging noise (line 141).

Are the hyperspectral images and other materials available in a public repository? I tried to open the link and it appeared to be private or broken.

The original upload was apparently incomplete but not reported by figshare. The upload has now been repeated and tested. The link (line 475) remains private but should be accessible to the reviewers. If the submission is accepted, it will be made public.

Figure 2 is nicely illustrative, and I wonder if one could be possible to present the full data in a somewhat analogous way. For example, why can't we have a scatter plot of the form as Figure 2 but for all images, e.g. randomly draw 10,000 pixels from across the 50 images, so we have a nice selection of colors, do it a bunch of times (i.e. different draws of 10,000

pixels), and you can see from the panels the relationship of the color-appearance shift with hue (and chroma)?

We experimented with individual scatter plots laid out in a multipanel array to match Fig. 2. About 2/3 of the plots were unimodal and 1/3 multimodal, but did not offer much more insight than the statistical analysis, which actually quantifies the explanatory strengths of chroma (43%) and hue (27%). We also experimented with a smaller composite scatterplot based on all the scenes, but it proved too cluttered.

Appendix B

RESPONSE TO REFEREE

Our responses are in blue.

Referee: 1

Comments to the Author(s).

The authors did an excellent job in revising the paper. But while it is all technically correct, I do struggle with the conclusion. If color constancy should fail in the real world, it will ALWAYS include a memory component.

Further studies of colour memory are now mentioned. In many cases, colour constancy judgements will indeed involve memory but there are also everyday colour constancy experiences where a cloud passes over the sun or a shadow is cast locally by an object, neither of which place any great load on memory. These examples are now considered in the Discussion (para. beginning “Second”).

We never see two visual worlds side by side.

But we do make side-by-side constancy judgements within or between scenes under different lights, e.g. surfaces partly in direct sunlight and partly in shade. Justification for using psychophysical data from side-by-side comparisons is now given in the Discussion (para. beginning “Second”).

Thus, a much higher delta E threshold would really be more appropriate, maybe twice as high as the ones used now. I would suggest the authors include such a higher value in their Figure 4. I suspect the conclusion might then be that failures are exceedingly rare.

Our choice of threshold was supported by published experimental data. The reviewer’s suggestion of doubling it seems arbitrary, but we have now inserted a sentence “Conversely, if thresholds were set larger or changes in illuminant spectrum made smaller, then these proportions decrease.” (Section 3(b)).

Some of the other points I made in my earlier review are dealt with now, but the answers seem to be well hidden.

We have now split the Discussion points into four separate items to make these answers clearer.

For example, how much of the predicted deviations could be due to the color appearance models used? This is a question many readers may ask.

We have used two models for estimating colour differences, both of which have been widely tested outside the laboratory, and one model based on cone-excitation ratios, which is physiological and not a model of colour appearance in any normal sense. All three models made similar predictions.

Nevertheless, we did add a caveat in the revision (Discussion, penultimate para.) where we allowed that colour differences may be less obvious in the real world. We also pointed out in the revision (Introduction, penultimate para.) why making real-world measurements is so difficult.